# Postoperative Bladder Neck to Pubic Symphysis Ratio Predictive for De Novo Overactive Bladder after Robot-Assisted Radical Prostatectomy

**DOI:** 10.3390/diagnostics13203173

**Published:** 2023-10-11

**Authors:** Nayuka Matsuyama, Taku Naiki, Shuzo Hamamoto, Yosuke Sugiyama, Yasue Kubota, Takashi Hamakawa, Toshiki Etani, Shoichiro Iwatsuki, Kazumi Taguchi, Yuya Ota, Masakazu Gonda, Maria Aoki, Toshiharu Morikawa, Taiki Kato, Atsushi Okada, Takahiro Yasui

**Affiliations:** 1Department of Nephro-Urology, Graduate School of Medical Sciences, Nagoya City University, Nagoya 467-8601, Japan; nayuka518.g@gmail.com (N.M.); hamamo10@med.nagoya-cu.ac.jp (S.H.); uroetani@med.nagoya-cu.ac.jp (T.E.); iwatsuki@med.nagoya-cu.ac.jp (S.I.); ktaguchi@med.nagoya-cu.ac.jp (K.T.); yuyaota0307@gmail.com (Y.O.); mgonda@med.nagoya-cu.ac.jp (M.G.); hd.maria@med.nagoya-cu.ac.jp (M.A.); t-mrkw@med.nagoya-cu.ac.jp (T.M.); a-okada@med.nagoya-cu.ac.jp (A.O.); yasui@med.nagoya-cu.ac.jp (T.Y.); 2Department of Pharmacy, Nagoya City University Hospital, Nagoya 467-8601, Japan; phsugi@med.nagoya-cu.ac.jp; 3Department of Clinical Physiology, Graduate School of Nursing, Nagoya City University, Nagoya 467-8601, Japan; yasuekbt@med.nagoya-cu.ac.jp; 4Department of Urology, Nagoya City University West Medical Center, Nagoya 462-8508, Japan; hamakawa@med.nagoya-cu.ac.jp; 5Department of Urology, Nagoya City University East Medical Center, Nagoya 464-8547, Japan; tkatouro@med.nagoya-cu.ac.jp

**Keywords:** de novo overactive bladder, postoperative bladder neck to pubic symphysis ratio, Retzius sparing

## Abstract

Background: The aim was to investigate the incidence and clinical predictive factors of de novo overactive bladder (OAB) after robot-assisted radical prostatectomy (RARP), including a Retzius-sparing (RS) approach, in the same period at a single institution. Methods: Of a total of 113 patients with localized prostate cancer, 81 received conventional RARP (CON-RARP) and 32 received RS-RARP at our institution. The basic characteristics data of patients and self-assessment questionnaires, including IPSS and OABSS, were obtained preoperatively and 1, 3, and 6 months after RARP. In addition, a retrospective biomarker analysis was also performed of predictive clinical parameters obtained from cystography that included a postoperative bladder neck to pubic symphysis (BNPS) ratio. Results: Patients’ basic characteristics were similar between CON-RARP and RS-RARP groups. With respect to the surgical procedure, anastomosing time was found to be significantly longer for patients in the RS-RARP compared to the CON-RARP group (*p* < 0.01). Compared to the CON-RARP group, the RS-RARP group showed a significantly lower postoperative BNPS and aspect ratio (*p* < 0.001). The incidence of de novo OAB in patients of the CON-RARP group was greater than for those in the RS-RARP group (40.7% CON-RARP vs. 25.0% RS-RARP), though this was not significant. Regarding the emergence of de novo OAB, the following were revealed in univariate analysis to be independent prognostic factors: age > 64 years (hazards ratio [HR]: 4.32, 95% confidence interval [CI]: 1.51–12.3), postoperative BNPS ratio > 0.44 (HR: 8.7, 95% CI: 6.43–54.5), postoperative aspect ratio > 1.18 (HR: 3.36, 95% CI: 1.49–7.61). Additionally, multivariate analysis identified a sole significant prognostic factor: postoperative BNPS ratio > 0.44 (HR: 13.3, 95% CI: 4.33–41.1). Conclusion: Our findings indicate that the postoperative BNPS ratio may be a practical predictive indicator of the emergence of de novo OAB after RARP.

## 1. Introduction

In patients with localized prostate cancer, a range of therapies can be addressed; however, above all, radical prostatectomy is a mainstay of treatment [1,2]. In addition, a common irritating adverse event that influences quality of life after prostatectomy is urinary incontinence [3,4]. Nowadays, robot-assisted radical prostatectomy (RARP) has become popular, and has undergone various improvements to control incontinence via bladder neck preservation, sparing the pubovesical complex, and restoration of the posterior aspect of the rhabdosphincter; it has shown better outcomes for urinary continence compared with previous methods [5]. However, the causes of urinary continence after prostatectomy remain unclear and methods to prevent post-operative incontinence remain unestablished. Recently, several studies described how de novo overactive bladder (OAB), without or with urinary incontinence, may also develop after prostatectomy. The incidence of OAB is thought to be underestimated. Only a few reports exist on long-term bladder dysfunction after prostatectomy, including OAB [6]. To date, robust data are lacking on the incidence and precise physiological mechanisms of de novo OAB after RARP.

Due to a recent improvement in surgical technique, Retzius space-sparing (RS) during RARP may greatly impact the urinary continence rate [7,8]. Several subsequent studies verified the efficacy of RS-RARP by preserving the Retzius space, arcus tendinous, neurovascular bundle, endopelvic fascia, and deep dorsal vein complex—important structures that maintain normal urinary continence [9,10,11]. However, no report describes the emergence of de novo OAB after prostatectomy, including in RS approaches. Therefore, in this study, we retrospectively analyzed the incidence of de novo OAB after RARP, including an RS approach, in the same period in our institution. We also explored potential predictive biomarkers in this cohort.

## 2. Materials and Methods

### 2.1. Inclusion Criteria for Patient Enrolment

Between October 2017 and November 2021, a total of 171 patients underwent RARP at our single institution. We prospectively collected and assessed pathological data, as well as incontinence outcomes. In addition to patients’ basic characteristics data, self-assessment questionnaires, including the international prostate symptom score and overactive bladder symptom score (OABSS), were obtained preoperatively and 1, 3, and 6 months after RARP. As shown in the flowchart in Figure 1, after excluding patients with lost follow-up, the existence of preoperative OAB, or with missing data, 113 patients were finally included in this study. These patients underwent CON-RARP (*n* = 81; CON-RARP group) or RS-RARP (*n* = 32; RS-RARP group) in the same period and were retrospectively analyzed. This study was undertaken with the approval of the ethics committees of all universities belonging to this group (approval no. 60-19-0096) and websites outlined opt-out information for patients. The design of the investigation was in accordance with the Declaration of Helsinki (2013).

### 2.2. Pre- and Post-Operative Assessment and Data Collection

The definition of an overactive bladder included having an urgency to urinate greater than once a week and ≥3 points according to OABSS. The surgical techniques involved in CON-RARP and RS-RARP have been previously described [10,11]. As described by Olgin et al. (2014) [12], cystography is performed approximately 7 days after RARP. Calculations of bladder neck to pubic symphysis (BNPS) ratios were made by a single observer (N. M.) from postoperative cystography and were based on the distance between the new bladder neck and pubic symphysis divided by the total height of the pubic symphysis (Figure 2a). And the aspect ratio was calculated using the vertical and horizontal length in cystography (Figure 2b). In addition, the preoperative membranous urethral length (MUL) was also evaluated by a single observer (N. M.) in preoperative magnetic resonance images (MRI) as previously described [11,13]. Furthermore, the medical records of patients at our institution were used to extract patient information on basic data including age, height, weight, prostate weight, clinicopathological data, and on perioperative data, including the estimated loss of bleeding and anastomosing times.

### 2.3. Statistics

To evaluate differences in categorical parameters, we used Fisher’s exact test or a Mann–Whitney *U* test. After a receiver operating characteristic (ROC) curve analysis of the total cohort with regard to the emergence of de novo OAB, new cut-off values of parameters were determined. We conducted both univariate and stepwise multivariate logistic regression analyses to compare parameters between control and de novo OAB groups. Variables that were clinically important factors were used to predict the emergence of de novo OAB. Data were evaluated with the use of EZR software (ver. 1.61) (Saitama Medical Center, Jichi Medical University, Yakushiji, Japan).

## 3. Results

### 3.1. Characteristics and Patient Outcomes by Retrospective Study

As shown in Table 1, patients’ basic characteristics were similar, including median prostate weight, median age, median body mass index (BMI), and the distribution of preoperative Gleason score. With regard to new cut-off values for the emergence of de novo OAB, clinical factors including BMI, estimated loss of bleeding, prostate weight, anastomosing time, preoperative MUL, postoperative BNPS ratio, and postoperative aspect ratio were analyzed using receiver operator characteristic curves. New cut-off values were determined to be a BMI of 24.5 kg/m^2^, an estimated blood loss of 507.0 mL, a prostate weight of 46.0 g, an anastomosing time of 31 min, a preoperative MUL of 11.1 mm, a postoperative BNPS ratio of 0.44, and a postoperative aspect ratio of 1.18, as shown in Figure 3. As shown in Table 1, reflecting the surgical procedure, compared with patients in the CON-RARP group, the anastomosing time was found to be significantly longer in a higher proportion of patients in the RS-RARP group (*p* < 0.01). Postoperative BNPS and aspect ratios were found to be significantly lower in a higher proportion of patients in the RS-RARP group compared to those in the CON-RARP group (*p* < 0.001). The incidence of de novo OAB in patients in the CON-RARP group was greater than that found in those in the RS-RARP group (40.7% CON-RARP vs. 25.0% RS-RARP), though this was not significant.

### 3.2. Identification of Prognostic Factors for the Emergence of De Novo OAB

Figure 4 shows the seven parameters subdivided according to patients without (control) and with de novo OAB (de novo) in each group. For median age, BMI, estimated blood loss, prostate weight, and preoperative MUL, significant differences between the four groups were not found (Figure 4a–d,f). The median anastomosing time was found to be significantly greater, and the postoperative aspect ratio of the RS-RARP group was found to be significantly lower than that in the CON-RARP group. However, when comparing control and de novo groups, these were similar (Figure 4e,h). Concerning the postoperative BNPS ratio, the median scores of the CON-RARP group were significantly greater than those of the RS-RARP group; when compared with the respective control, they were also significantly greater in the de novo groups (Figure 4g; *p* < 0.0001). In using new cut-off values of ROC, prognostic factors for de novo OAB were identified from univariate and multivariate analyses of baseline and perioperative parameters (Table 2). Regarding the emergence of de novo OAB, the following were revealed in univariate analysis to be independent prognostic factors: age > 64 years (HR: 4.32, 95% CI: 1.51–12.3), postoperative BNPS ratio > 0.44 (HR: 8.7, 95% CI: 6.43–54.5), postoperative aspect ratio > 1.18 (HR: 3.36, 95% CI: 1.49–7.61). Additionally, multivariate analysis identified as a sole significant prognostic factor a postoperative BNPS ratio > 0.44 (HR: 13.3, 95% CI: 4.33–41.1) (Table 2).

## 4. Discussion

In this study on Japanese patients with localized prostate cancer, we studied the development of de novo OAB after RARP, including a Retzius-sparing approach. Several studies have described the efficacy, in particular in immediate continence after surgery, of an RS approach [7,9,10,11]. A recent up-to-date and comprehensive meta-analysis revealed the RS-RARP group as having a significantly improved postoperative continence rate when compared to the CON-RARP group [14]. Additionally, the RS-RARP group showed no significant differences in estimated blood loss but a shorter console time. However, no report exists with detailed information as to the development of de novo OAB after RS-RARP. The pathophysiology of OAB is multifactorial, and, therefore, it is difficult to explain the basis for de novo OAB developing after RARP in a coherent manner [15]. Of such factors, the most well-known obtained by urodynamic findings is detrusor overactivity (DO). Previous reports described de novo DO ranging from 2–77% after radical prostatectomy [3]. Ficazzola et al. (1998) [16] described how 89% of patients with incontinence after radical prostatectomy presented with DO and decreased bladder capacity may lead to the emergence of de novo OAB [3,17,18]. Detrusor overactivity following radical prostatectomy has predominantly been caused by partial denervation of the bladder in surgery. Furthermore, however, recent reports described another hypothesis in that neural mechanisms induced by urethral afferent fibers may cause bladder contraction leading to OAB symptoms [19,20]. In the CON-RARP procedure, a clear operative field can be obtained via the dissection of the Retzius space. However, this can result in injury to important structures and nerves of the bladder associated with urine continence, such as the neurovascular bundle, endopelvic fascia, and feeding vessels. In comparison, in RS-RARP, not only the Retzius space but also the pelvic floor muscle can be preserved; surgical trauma is minimized, and normal anatomical structures are retained as much as possible [9,10]. In our study, in the total cohort, the incidence of de novo OAB was 36.3% (41/113); this was lower in the RS-RARP compared to CON-RARP group but was not statistically significant (40.7% CON-RARP vs. 25.0% RS-RARP). This result could be partly due to maximization of the preservation of natural anatomy in an RS approach. The accumulation of further clinical cases is required to recognize the superiority of de novo OAB. A bigger prospective intervention study would resolve these questions.

De novo OAB after RARP can be a burden in patients. The prediction of this potentially overlooked population is important since multiple new therapies are currently available for OAB [21]. To date, several reports exist on the biological or operative factors that predict urinary incontinence following radical prostatectomy. The etiology of postoperative incontinence, anatomic supportive structures, and innervation in the pelvis appeared to be important factors. An older age at surgery, high BMI, and pre-existing lower-urinary-tract symptoms negatively affect continence [4,18,22]. In terms of MUL, several studies reported the correlation between preoperative and/or postoperative MUL and incontinence after radical prostatectomy [13,23]. Above all, in preoperative MUL, the impact was controversial. Several reports described how a longer preoperative MUL was greatly correlated to recovery from incontinence. However, after an analysis of 50 consecutive patients who underwent preoperative and postoperative MRI, Haga et al. (2014) [24] reported a significant association of postoperative MUL with both continence grade and the quality of life (QOL) index. However, a significant association was not found between preoperative MUL and either continence grade or the QOL index. In addition, the postoperative BNPS ratio was described as a predictor of early continence recovery after RARP; however, as a convenient predictive biomarker of the development of de novo OAB following RARP, this was very limited. Koguchi et al. (2019) [25] showed that by using factors evaluated by a preoperative cardio-ankle vascular index, a significant increase was noted in the prevalence of de novo OAB in an atherosclerosis group 3 months after RARP compared to a group without atherosclerosis. A recent retrospective analysis found that BMI was an independent factor in the emergence of de novo OAB 6 months following RARP [26]. In our study, several baseline preoperative and postoperative parameters, including measures obtained by MRI and cystography, were examined. After setting the new cut-off value by the ROC curve, only a postoperative BNPS ratio >0.44 was a very strong independent risk factor for the emergence of de novo OAB post-RARP in univariate and multivariate analyses. Additionally, the RS-RARP group showed a significantly lower postoperative BNPS ratio compared to that of the CON-RARP group. Furthermore, when compared with the control, the postoperative BNPS ratio was also significantly greater in both de novo CON-RARP and RS-RARP groups. Our study was the first to describe the positive effect of the postoperative BNPS ratio and negative effect of the preoperative MUL for predicting de novo OAB after RARP, including for an RS approach. The mechanism for the effect of a lower postoperative BNPS ratio in physiological attenuation during the emergence of OAB is unclear. However, in future, when accumulating data for de novo OAB in post-prostatectomy patients, any therapeutic decision and early intervention using the biomarker in cystography can be individualized for each patient.

Our study has several limitations. It is a retrospective analysis and therefore has shortcomings, such as relatively small sample sizes that may lead to non-significance in the incidence between CON-RARP and RS-RARP groups. The selection of surgical procedure depends on the clinician’s preference; therefore, the setting of statistical power in advance was not performed. A post hoc statistical power of 29.6% was obtained for the incidence of de novo OAB between each group using a two-group t-test with a two-sided significance level of *p* < 0.05. Second, we could not analyze bladder function since data on DO were lacking in our study. Finally, the period of follow-up was relatively short. A long-term follow-up investigation to support the conclusions of our study is warranted.

## Figures and Tables

**Figure 1 diagnostics-13-03173-f001:**
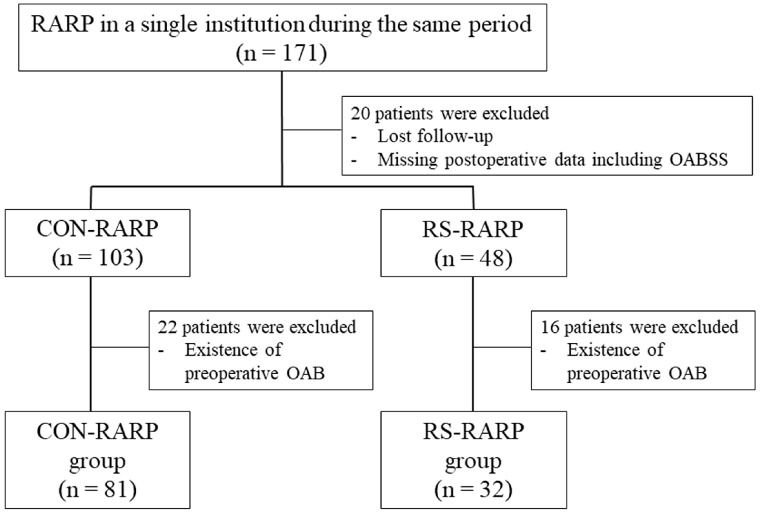
Study design for this retrospective cohort.

**Figure 2 diagnostics-13-03173-f002:**
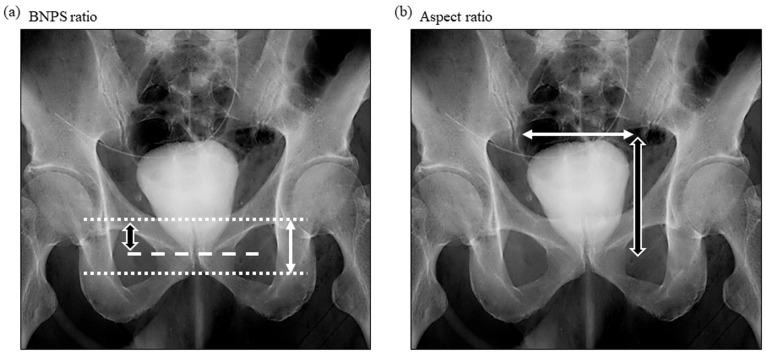
Illustration of BNPS ratio and aspect ratio. Using the cystography obtained after RARP, BNPS ratio was used for the biomarker analysis. BNPS ratio was calculated by dividing the distance between the new bladder neck and the pubic symphysis (white and black arrow) by the total pubic symphysis length (whited arrow). Aspect ratio was calculated by the vertical (white and black arrow) and horizontal (white arrow) length. BNPS ratio: bladder neck to pubic symphysis ratio; RARP: robot-assisted radical prostatectomy.

**Figure 3 diagnostics-13-03173-f003:**
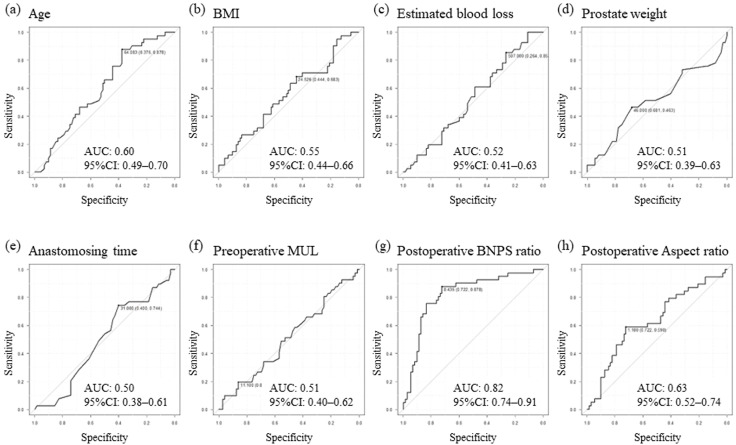
The setting of cut-off values using ROC curve analysis of indicators in the total cohort. Age (**a**), BMI (**b**), estimated blood loss (**c**), prostate weight (**d**), anastomosing time (**e**), preoperative MUL (**f**), postoperative BNP ratio (**g**), and postoperative aspect ratio (**h**). AUC: area under the curve; BMI: body mass index; BNPS ratio: bladder neck to pubic symphysis ratio; CI: confidence interval; MUL: membranous urethral length; ROC: receiver operating characteristic.

**Figure 4 diagnostics-13-03173-f004:**
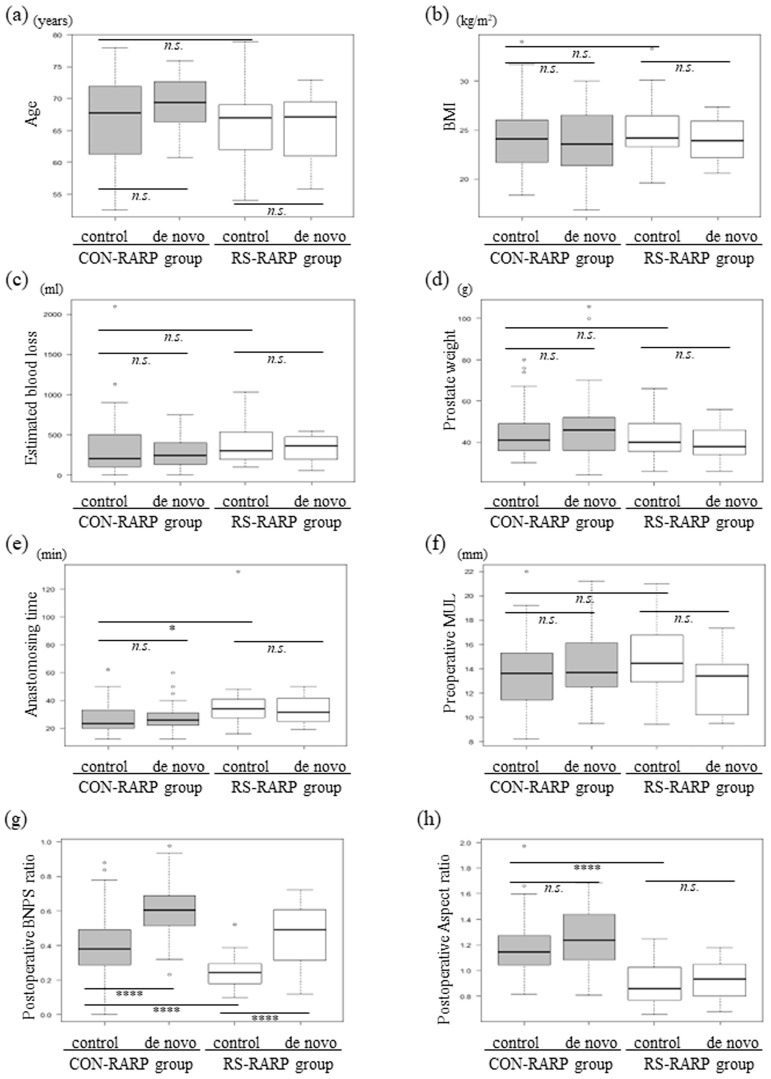
Distribution of clinical or intra-postoperative parameters divided into de novo OAB or non-OAB (control) groups in each patient group. Age (**a**), BMI (**b**), estimated blood loss (**c**), prostate weight (**d**), anastomosing time (**e**), preoperative MUL (**f**), postoperative BNP ratio (**g**), and postoperative aspect ratio (**h**). BMI: body mass index; BNPS ratio: bladder neck to pubic symphysis ratio; CON-RARP: conventional RARP; MUL: membranous urethral length; OAB: overactive bladder; RARP: robot-assisted radical prostatectomy; RS-RARP: Retzius-sparing RARP. * *p* < 0.05, **** *p* < 0.0001, n.s., not significant.

**Table 1 diagnostics-13-03173-t001:** Patients’ characteristics and outcomes of CON-RARP and RS-RARP in a retrospective cohort.

Characteristics		CON-RARP Group (*n* = 81)	RS-RARP Group (*n* = 32)	*p* Value
Median age, years (range)		68 (53–78)	67 (54–79)	0.16
Median BMI, kg/m^2^ (range)		23.9 (16.8–34.0)	24.0 (19.6–33.3)	0.30
Median prostate weight, g (range)		42.0 (24.0–106.0)	39.0 (26.0–66.0)	0.20
ISUP grade group, *n* (%)	1	11 (13.6)	6 (18.8)	0.24
	2	41 (50.6)	10 (31.2)	
	3	25 (30.9)	14 (43.8)	
	4	2 (2.5)	2 (6.2)	
	5	2 (2.5)	0 (0)	
BMI, *n* (%)	≤24.5 kg/m^2^	49 (60.5)	19 (59.4)	1
	>24.5 kg/m^2^	32 (39.5)	13 (40.6)	
Estimated blood loss, *n* (%)	≤507.0 mL	63 (77.8)	24 (75.0)	0.81
	>507.0 mL	18 (22.2)	8 (25.0)	
Prostate weight, *n* (%)	≤46.0 g	48 (59.3)	23 (71.9)	0.28
	>46.0 g	33 (40.7)	9 (28.1)	
Anastomosing time, *n* (%)	≤31.0 min	54 (70.1)	12 (37.5)	<0.01 **
	>31.0 min	23 (29.9)	20 (62.5)	
Preoperative MUL, *n* (%)	≤11.1 mm	12 (14.8)	5 (15.6)	1
	>11.1 mm	69 (85.2)	27 (84.4)	
Postoperative BNPS ratio, *n* (%)	≤0.44	31 (38.3)	26 (81.2)	<0.001 ***
	>0.44	50 (61.7)	6 (18.8)	
Postoperative Aspect ratio, *n* (%)	≤1.18	39 (49.4)	30 (93.8)	<0.001 ***
	>1.18	40 (50.6)	2 (6.2)	
The incidence of de novo OAB, *n* (%)		33 (40.7)	8 (25.0)	0.13

BMI: body mass index; BNPS ratio: bladder neck to pubic symphysis ratio; CON-RARP: conventional RARP; ISUP: International Society of Urological Pathology; MUL: membranous urethral length; OAB: overactive bladder; RARP: robot-assisted radical prostatectomy; RS-RARP: Retzius-sparing RARP ** *p* < 0.01, *** *p* < 0.001.

**Table 2 diagnostics-13-03173-t002:** Univariate and multivariate analyses of baseline and postoperative parameters, and incidence of de novo OAB in 113 patients of the total cohort.

Parameters	Univariate			Multivariate		
	OR	95% CI	*p* Value	OR	95% CI	*p* Value
Age, >64 vs. ≤64	4.32	1.51–12.3	<0.01 **	2.73	0.80–9.3	0.11
BMI, >24.5 vs. ≤24.5	0.58	0.26–1.3	0.19	-	-	-
The choice of surgical approach, Retzius-sparing vs. conventional	0.49	0.19–1.21	0.12	-	-	-
Estimated loss of bleeding, >507 vs. ≤507	0.57	0.22–1.51	0.26	-	-	-
Prostate weight, >46.0 vs. ≤46.0	1.84	0.84–450	0.13	-	-	-
Anastomosing time, >31.0 vs. ≤31.0	0.67	0.30–1.51	0.33	-	-	-
Preoperative MUL, >11.1 vs. ≤11.1	0.78	0.27–2.24	0.65	-	-	-
Postoperative BNPS ratio, >0.44 vs. ≤0.44	8.7	6.43–54.5	<0.0001 ****	13.3	4.33–41.1	<0.0001 ****
Postoperative Aspect ratio, >1.18 vs. ≤1.18	3.36	1.49–7.61	<0.01 **	1.56	0.57–4.26	0.38

BMI: body mass index; BNPS ratio: bladder neck to pubic symphysis ratio; CI: confidence interval; OR: odds ratio; MUL: membranous urethral length; RARP: robot-assisted radical prostatectomy; CON-RARP: conventional RARP; RS-RARP: Retzius-sparing RARP; OAB: overactive bladder. ** *p* < 0.01, **** *p* < 0.0001.

## Data Availability

Materials and raw data may be given after a request made to the corresponding author.

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
