# Peer review of "Postoperative Bladder Neck to Pubic Symphysis Ratio Predictive for De Novo Overactive Bladder after Robot-Assisted Radical Prostatectomy"

_diagnostics, 2023, doi:10.3390/diagnostics13203173_

Round 1

Reviewer 1 Report

1)     This report is interesting. However, the small number of cases may make it difficult to draw definitive conclusions.

2)     The authors had better to provide specific illustrations from the bladder neck to the pubic symphysis (BNPS).

3)     The authors need to describe the definition of Aspect ratio in the method.

4)     In Table 1, the results for the Postoperative BNPS ratio and Postoperative Aspect ratio are reversed.

Author Response

October 1st, 2023

Prof. Andreas Kjaer

Editor-in-Chief,

Diagnostics

Dear Prof,

We would like to submit our manuscript entitled “Postoperative bladder neck to pubic symphysis ratio predictive for de novo overactive bladder after robot-assisted radical prostatectomy” to be carefully assessed by reviewers. We have studied the reviewers’ comments and have revised the manuscript accordingly. Consequently, it is my pleasure to resubmit the revised version to Diagnostics.

We would like to thank the reviewers for their helpful comments, which gave us a better perspective on our work. We have revised the manuscript in line with the reviewers’ suggestions, with all changes made in red text.

The attached paper has been carefully reviewed by an experienced medical editor whose first language is English and who is specialized in the editing of papers written by physicians and scientists whose native language is not English. We strongly believe that the extensive revisions have substantially improved our manuscript and hope that it is now suitable for publication in Diagnostics.

Please address questions and any correspondence to:

Taku Naiki M.D. Ph.D.   

Department of Nephro-urology, Nagoya City University, Graduate School of Medical Sciences, Kawasumi 1, Mizuho-cho, Mizuho-ku 467-8601, Nagoya, Japan.

Tel: 81-52-853-8266

Fax: 81-52-852-3179

E-mail: rx-nike@hotmail.co.jp, naiki@med.nagoya-cu.ac.jp

Responses to Reviewers

Reviewer 1

  1. This report is interesting. However, the small number of cases may make it difficult to draw definitive conclusions.

In response to the reviewer’s comment, we added the sentence as limitation as below.

“It is a retrospective analysis and therefore had shortcomings such as relatively small sample sizes that may lead to non-significance in the incidence between CON-RARP and RS-RARP groups. the selection of surgical procedure depends on the clinician’s preference, therefore, the setting of statistical power in advance was not performed. A post hoc statistical power of 29.6% was obtained for the incidence of de novo OAB between each group using a two-group t-test with a two-sided significance level of p < 0.05.” in Discussion

  1. The authors had better to provide specific illustrations from the bladder neck to the pubic symphysis (BNPS).

In response to the reviewer’s recommendation, we newly added the illustration of BNPS ratio and Aspect ratio as Figure 2, and modified the main manuscript.

  1. The authors need to describe the definition of Aspect ratio in the method.

In response to the reviewer’s comments, we added the definition of Aspect ratio in Methods.

  1. In Table 1, the results for the Postoperative BNPS ratio and Postoperative Aspect ratio are reversed.

In response to the reviewer’s designation, we modified the results in Table 1.

--------------------------------------------------------------------

Reviewer 2 Report

the authors compred two RARP approaches in terms of de novo OAB occurence. Morevover, this study assese new predictor of OAB, namely bladder neck to pubic symphysis (BNPS) ratios.

Overall, the study sounds scientifically well. I have just a minor suggestion. 

- in the method, plòease clarify how the patients were assigned to the study arms. Was this study a randomized ? Furthermore, the statisticalpower of the study, should be reported in the statistical method being this a comparative study. 

none

Author Response

October 1st, 2023

Prof. Andreas Kjaer

Editor-in-Chief,

Diagnostics

Dear Prof,

We would like to submit our manuscript entitled “Postoperative bladder neck to pubic symphysis ratio predictive for de novo overactive bladder after robot-assisted radical prostatectomy” to be carefully assessed by reviewers. We have studied the reviewers’ comments and have revised the manuscript accordingly. Consequently, it is my pleasure to resubmit the revised version to Diagnostics.

We would like to thank the reviewers for their helpful comments, which gave us a better perspective on our work. We have revised the manuscript in line with the reviewers’ suggestions, with all changes made in red text.

The attached paper has been carefully reviewed by an experienced medical editor whose first language is English and who is specialized in the editing of papers written by physicians and scientists whose native language is not English. We strongly believe that the extensive revisions have substantially improved our manuscript and hope that it is now suitable for publication in Diagnostics.

Please address questions and any correspondence to:

Taku Naiki M.D. Ph.D.   

Department of Nephro-urology, Nagoya City University, Graduate School of Medical Sciences, Kawasumi 1, Mizuho-cho, Mizuho-ku 467-8601, Nagoya, Japan.

Tel: 81-52-853-8266

Fax: 81-52-852-3179

E-mail: rx-nike@hotmail.co.jp, naiki@med.nagoya-cu.ac.jp

Responses to Reviewers

Reviewer 2

The authors compared two RARP approaches in terms of de novo OAB occurrence. Moreover, this study assesses new predictor of OAB, namely bladder neck to pubic symphysis (BNPS) ratios. Overall, the study sounds scientifically well. I have just a minor suggestion. 

  1. - in the method, please clarify how the patients were assigned to the study arms. Was this study a randomized? Furthermore, the statistical power of the study, should be reported in the statistical method being this a comparative study. 

This analysis is based on the retrospective design of the same period in single institution, the selection of surgical procedure depends on the clinician’s preference, therefore, the setting of statistical power in advance was not performed. A post hoc statistical power of 29.6% was obtained for the incidence of de novo OAB between each group using a two-group t-test with a two-sided significance level of p < 0.05, therefore, we added the sentence as limitation in Discussion as below.

“It is a retrospective analysis and therefore had shortcomings such as relatively small sample sizes that may lead to non-significance in the incidence between CON-RARP and RS-RARP groups. the selection of surgical procedure depends on the clinician’s preference, therefore, the setting of statistical power in advance was not performed. A post hoc statistical power of 29.6% was obtained for the incidence of de novo OAB between each group using a two-group t-test with a two-sided significance level of p < 0.05.” in Discussion

==========================================

Round 2

Reviewer 1 Report

The authors have successfully revised the previous manuscript.

Author Response

Thank you for the time spent in reviewing our article.